# Pediatric Osteoarticular *Kingella kingae* Infections of the Hand and Wrist: A Retrospective Study

**DOI:** 10.3390/microorganisms11082123

**Published:** 2023-08-21

**Authors:** Blaise Cochard, Elvin Gurbanov, Ludmilla Bazin, Giacomo De Marco, Oscar Vazquez, Giorgio Di Laura Frattura, Christina N. Steiger, Romain Dayer, Dimitri Ceroni

**Affiliations:** 1Pediatric Orthopedics Unit, Pediatric Surgery Service, Geneva University Hospitals, CH-1211 Geneva, Switzerland; blaise.cochard@hcuge.ch (B.C.); oscar.vazquez@hcuge.ch (O.V.); christina.steiger@hcuge.ch (C.N.S.); romain.dayer@hcuge.ch (R.D.); 2Division of Orthopedics and Trauma Surgery, Geneva University Hospitals, CH-1211 Geneva, Switzerland; elvin.gurbanov@hcuge.ch (E.G.); ludmilla.bazin@hcuge.ch (L.B.)

**Keywords:** osteoarticular infections, *Kingella kingae*, hand, wrist

## Abstract

Our understanding of pediatric osteoarticular infections (OAIs) has improved significantly in recent decades. *Kingella kingae* is now recognized as the most common pathogen responsible for OAIs in pediatric populations younger than 4 years old. Research has provided a better understanding of the specific types, clinical characteristics, biological repercussions, and functional outcomes of these infections. Hands and wrists are rarely infected, with few reports available in the literature. The present study aimed to examine this specific condition in a large patient cohort, explore the implications for each anatomical area using magnetic resonance imaging (MRI), and critically evaluate the evolution of therapeutic management.

## 1. Introduction

In the last three decades, the accelerated development and widespread use of new screening tests have simplified the identification of different bacteria and improved the detection of osteoarticular infections (OAIs). Indeed, the use of nucleic acid amplification assays has provided irrefutable evidence that *Kingella kingae* (*K. kingae)* is today the most common pathogen responsible for OAIs among children younger than 4 years old [1,2,3,4]. The easier recognition of *K. kingae* infections has enabled a more accurate understanding of their specific types, clinical characteristics, and biological repercussions.

Septic arthritis is very likely the most common osteoarticular disease due to *K. kingae*. Reports have suggested that it constitutes between 53% and 82.8% of all OAIs due to this pathogen [5,6,7]. *K. kingae* usually affects large joints of the upper and lower extremities such as the hip, knee, ankle, shoulder, or elbow joint [6,7,8,9,10,11,12,13], with the knee involved most frequently. *K. kingae* osteomyelitis primarily affects long bones, like the femur, tibia, humerus, radius, and ulna [1,2,3,4,14,15]. It is noteworthy that the epiphysis or apophysis, which are almost never invaded by other organisms, are commonly affected by *K. kingae* osteomyelitis [16,17,18]. Invasive *K. kingae* infections can also lead to atypical osteoarticular infections, such as spondylodiscitis, pyomyositis, cellulitis, bursitis, and tendon sheath infections [12,19,20]. Atypical locations, such as the sternoclavicular, acromioclavicular, tarsal, or metacarpophalangeal/metatarsophalangeal joints, and unusual bones, like the talus, sternum, or clavicle, are more frequently found in *K. kingae* OAIs than in OAIs due to other pathogens [6,7,8,9,10,11,12,13,21]. Attending physicians must, therefore, remain alert when consulting small children presenting with osteoarticular symptoms suggestive of an OAI in unconventional anatomical sites.

Few reports in the medical literature have described hand or wrist infections due to *K. kingae* [19,22,23]. We were recently surprised to see how often *K. kingae* affected hands and wrists, suggesting this microorganism’s predisposition to infecting these atypical locations. However, we were also challenged by the heterogeneity of infections affecting hands and wrists since we noted that one or more anatomical compartments (bones, joints, tendon sheaths, subcutaneous tissues) can be involved. We, therefore, wished to estimate the true frequency of osteoarticular infections of the hand or wrist due to *K. kingae*. The present study aimed to examine OAIs of the hands and wrists caused by *K. kingae* in a large cohort of patients. We also wanted to explore the implications of *K. kingae* infections of the hands or wrists based on magnetic resonance imaging (MRI) scans. We thus sought to summarize and critically examine the different types of lesions encountered in OAIs of the hand and wrist due to *K. kingae*. Current trends in the therapeutic management of these lesions were also critically analyzed.

## 2. Materials and Methods

After approval from the Children’s Hospital Ethics Review Committee (CE 2023-102R), we retrospectively reviewed the medical charts of every child admitted to our institution between January 2007 and May 2023 for an OAI caused by *K. kingae* (January 2007 corresponds to the introduction of a routine molecular detection method for *K. kingae*). Our 111-bed tertiary pediatric hospital serves the city of Geneva and surrounding areas; it is the only facility providing inpatient and specialized medical services for pediatric OAIs to 460,000 local inhabitants. Diagnostic codes for septic arthritis, osteomyelitis (acute and subacute), spondylodiscitis, pyomyositis, tenosynovitis, and chondritis were used to identify the study population from our institution’s electronic patient records.

The criteria established by Morrey [24,25,26] and Morrissey [27] were used to estimate children’s risks of having a bone or joint infection. Diagnoses of an OAI were confirmed using MRI, according to established criteria [28]. Children diagnosed with a musculoskeletal infection were further categorized using the following diagnoses: septic arthritis, acute osteomyelitis, subacute osteomyelitis, osteomyelitis with concomitant arthritis, spondylodiscitis, septic myositis, septic chondritis, and septic tenosynovitis. Among these children, we selected those who had presented with an osteoarticular *K. kingae* infection selectively affecting their hand or wrist.

Information on age, sex, temperature at admission, the bone, joint, or tendon involved, and laboratory data, including bacterial cultures (blood, synovial fluid, and bone exudate), quantitative polymerase chain reaction (qPCR) assays [29], white blood cell (WBC) and differential platelet counts, erythrocyte sedimentation rate (ESR), and serum C-reactive protein (CRP), were all collected for analysis. We used the classic cut-off values for the following four variables, which are considered to have predictive value for infection parameters in clinical practice: fever, defined as an oral temperature of ≥38 °C; WBC count > 17,500/mm^3^ in children younger than 12 months old, >17,000/mm^3^ in children 13–24 months old, >14,500/mm^3^ in children 25–48 months old, and >12,000/mm^3^ in older children; CRP > 10 mg/L; and ESR > 20 mm/h. We also evaluated our results considering the cut-off values of the four main clinical and biological predictors for an OAI described in the Kocher and Caird algorithms: fever > 38.5 °C; WBC > 12.000/mm^3^; CRP ≥ 20 mg/L; and ESR ≥ 40 mm/h [30,31,32,33]. An elevated platelet count is not currently recognized as a diagnostic marker of an OAI, despite a platelet count of >361,500/mm^3^ having been integrated into a model for predicting a *K. kingae* OAI [1]. Thus, platelet count was evaluated using the absolute value beyond which it is considered that there is thrombocytosis (>392.000/mm^3^), but also taking into consideration the threshold limit (>361,500/mm^3^), which is more suggestive of an OAI caused by *K. kingae.*

Because many of our patients did not undergo invasive bacteriological investigations, few were considered to be confirmed cases of an OAI due to *K. kingae*—they corresponded to cases for which there were positive imaging studies (namely MRI) and pathogens had been isolated from their blood, bone, joint, or tendon sheath fluid by means of cultures or PCR assays. We considered it highly probable that children had an OAI due to *K. kingae* when their clinical presentation and imaging studies were positive and PCR assays of oropharyngeal specimens were positive for *K. kingae* too. Patients presumed to have an OAI due to *K. kingae* but who only had suggestive clinical and laboratory data to support their positive imaging results were excluded from the study.

### 2.1. Microbiological Methods

Our institution has systematically used blood cultures to isolate the microorganisms responsible for OAIs. The blood culture media used in the present study were BACTEC 9000 for the period before 2009 and an automated blood culture system (BD BACTEC FX) after that. Once collected, joint or tendon sheath fluids and bone aspirates were sent to the laboratory for Gram staining, cell counts, and immediate inoculation onto Columbia blood agar (incubated under anaerobic conditions), CDC anaerobe 5% sheep blood agar (incubated under anaerobic conditions), chocolate agar (incubated in a CO_2_-enriched atmosphere), and brain-heart broth. These media were incubated for 10 days. Two PCR assays were also used for bacterial identification when standard cultures were negative. Initial aliquots (100–200 µL) were stored at −80 °C until processing for DNA extraction. A universal, broad-range PCR amplification of the 16S rRNA gene was performed using BAK11w, BAK2, and BAK533r primers (Eurogentec, Seraing, Belgium). Since 2007, a real-time PCR assay was available to target the *K. kingae* gene’s rtx toxin [29]. This is designed to detect two independent genes from the *K. kingae* rtx toxin locus, namely *rtxA* and *rtxB* [29], and it was used to analyze different biological samples, such as synovial fluid, bone or discal biopsy specimens, or peripheral blood. Since September 2009, our institution has also carried out oropharyngeal swab PCRs for children from 6 months to 4 years old. It has been demonstrated that this simple technique for detecting *K. kingae* rtx toxin genes in the oropharynx provides strong evidence that this microorganism is responsible for OAIs in symptomatic children, or even stronger evidence that it is not [29].

### 2.2. Statistical Analysis

The characteristics of patients with an OAI caused by *K. kingae* were analyzed, and clinical manifestations and laboratory test results were expressed as medians and ranges or means and standard deviations. All statistical analyses were performed using Jamovi software, version 2.3 (The Jamovi project (2022), accessed from https://www.jamovi.org on 13 May 2023). Normality was assessed using a Shapiro–Wilk test. Variables resulting from a normal distribution are presented as mean values and standard deviations. Non-normally distributed variables are presented as median values.

## 3. Results

The medical files of 497 children diagnosed with an OAI were analyzed; 313 of them were from 6–48 months old. We identified 249 children with OAIs attributed to *K. kingae,* and involvement of the hand and/or wrist was observed in 33 cases. *K. kingae*’s causal responsibility was proved in 14 OAIs (42.4%), whereas the remaining 19 cases (57.6%) were considered highly probably caused by *K. kingae*. OAIs of the hand or wrist due to *K. kingae* affected female patients in 57.6% of cases, and the mean patient age (±SD) was 16.9 ± 9.4 months, ranging from 7–44 months old (Table 1).

The types of OAIs observed are listed in Table 1 and Table 2. The most common conditions encountered were septic arthritis (69.7% of cases) and tenosynovitis (51.2%) (Table 2; Figure 1). Furthermore, 42.4% of infections were present in a single compartment, with 57.6% involving multiple compartments. Among these, bone involvement was present in 30.3% of cases, joints were involved in 69.7%, and soft tissues were also in 69.7% (Table 3). The identification of *K.* kingae was confirmed via blood culture (10%), in a bone or fluid culture (13.3%), in a bone or joint fluid PCR assay specific for *K.* kingae (92.9%), or via positive PCR assays performed using oropharyngeal swabs (93.8%).

The distributions of the clinical and laboratory parameters are summarized in Table 4. Distribution was normal for temperature at admission (*p* = 0.08); WBC (*p* = 0.26); PLT (*p* = 0.078) and ESR (*p* = 0.078). Distribution was not normal for CRP (*p* < 0.001) and left shift (*p* < 0.001). We observed that 75.8% of patients with a *K. kingae* OAI were afebrile (T < 38 °C) at admission, and only 15.2% had a temperature above 38.5 °C during their clinical examination at admission. The mean WBC value was 12,500/mm^3^ ± 4370/mm^3^. WBC counts were considered elevated, according to normal, age-related cut-offs, in 18.2% of cases (>17,500/mm^3^ in children 6–12 months old, >17,000/mm^3^ in children 13–24 months old, and >14,500/mm^3^ in children 25–48 months old), whereas left shift was only noted in one patient with concomitant herpangina. On the other hand, WBC counts were superior to the threshold of >12,000/mm^3^ described by Kocher et al. [30] in 45.5% of children. The median CRP value was 20.0 mg/L, and CRP was considered abnormal (>10 mg/L) in 71.9% of *K. kingae*-positive patients. Only 50.0% of patients with an OAI caused by *K. kingae* had a CRP value above the limit considered to be a predictor of infection by Caird et al. (CRP ≥ 20 mg/L) [33]. The mean ESR reached 40.9 ± 19.2 mm/h and was considered abnormal (>20 mm/h) in 80.8% of patients. Only 46.2% of patients with an OAI caused by *K. kingae* had an ESR above the limit considered to be a predictor for septic arthritis by Kocher et al. (ESR ≥ 40 mm/h). 

Finally, the mean platelet count was 401,000/mm^3^ ± 131,000/mm^3^. Platelet counts were considered elevated (>392,000/mm^3^) in 45.5% of cases. However, when applying the threshold limit (>361,500/mm^3^) suggestive of an OAI caused by *K. kingae* [1], we noted that 57.6% of our patients’ values were considered abnormal.

Regarding the therapeutic management of these patients, 54.5% underwent medical management with an appropriate antibiotic therapy, whereas 45.5% underwent surgical treatment. Among septic arthritis and osteomyelitis, 52.2% and 40%, respectively, underwent medical management. Nevertheless, we noted a trend towards purely medical management of our patients in the last few years (Figure 2). Interestingly, clinical results (mobility, grasping, swelling, and pain) at 4 weeks after the start of treatment were favorable, whether management was medical or surgical. Indeed, mobility and grasping were restored in all our subjects. In the majority of cases, patients no longer had pain during mobilization and swelling had disappeared (88.5% and 96.2%, respectively).

## 4. Discussion

Hand infections are a significant cause of morbidity among infants. Hands usually undergo trauma when children fall or use them to explore the world around them, including their own mouths or animals’ mouths. Indeed, it is surprising that children do not develop hand infections more frequently. Because of their exposure to the elements, hands are usually infected secondary to trauma or skin lesions, and infections include paronychia, felons, and flexor tenosynovitis. In addition to these secondary infections, there are also infections due to the hematogenous spread of germs with an oropharyngeal starting point. Indeed, there is some evidence to suggest that most invasive infections in young children are caused by pathogens carried asymptomatically in the respiratory tract. Microorganisms residing in the mucosal surface, such as *Streptococcus pneumoniae*, *H. influenzae type b*, *Neisseria meningitidis,* or *K. kingae*, have the potential to penetrate the bloodstream, disseminate, and invade distant organs. Colonization of the respiratory tract by these organisms is, therefore, a prerequisite for later invasion [7,10]. Little work has focused on hand infections in small children, despite *K. kingae’s* responsibility in pyogenic tenosynovitis having been reported repeatedly [19,22,23]. The present study, therefore, reported on the largest published series of hand infections in children under 4 years of age and for whom *K. kingae* was strongly suspected as the causative germ.

Our findings confirmed the idea that OAIs of the hand or wrist caused by *K. kingae* essentially occur in children under 4 years of age. There are two reasons why it is no coincidence that these infections occur during this specific period in a child’s life. Firstly, it is the period of maximal oropharyngeal colonization by *K. kingae* [12,34,35,36,37,38,39,40]. Secondly, it has also been demonstrated that the maternal immunity conveyed to the fetus during pregnancy diminishes gradually during this period, above all from 6–24 months. Longitudinal investigations have proved that average IgG levels against *K. kingae* are elevated at birth and then slowly decrease, reaching their lowest point at 6–7 months postnatally [14,15]. Nevertheless, IgG levels remain low until 18–24 months old, at which point a progressive increase in the serum levels of the immunoglobulin is noted [14,15]. Thus, as with all bone and joint infections due to *K. kingae*, infants are more prone to sustaining hand infections caused by this pathogen between 6 and 48 months old.

Surprisingly, the present work also revealed that hands and wrists are frequently affected by *K. kingae* infections, constituting 13.3% of the total number of OAIs in our series. Seeing how these anatomical sites seem to be commonly infected by this specific germ in this age group, it appears that hand and wrist infections due to *K. kingae* have probably been underestimated in the past. Our results also demonstrated that most hand and wrist infections due to *K. kingae* affected several anatomical compartments and that arthritis and tenosynovitis were the most frequently represented lesions. Indeed, in 57.6% of our reported cases, infection affected two anatomical compartments, and arthritis was the most frequently encountered lesion.

The present study also demonstrated that hand infections caused by *K. kingae* infrequently generate fever: the mean temperature at admission was 37.3 °C. Even more interesting is the fact that only 12.9% of children with an OAI of the hand due to *K. kingae* presented with a temperature above 38.5 °C, an element that several authors believe is an essential predictor of an OAI. Analogously, a study of 11 pediatric patients with culture-proven tenosynovitis by Lironi et al. reported that only 27.3% of these children presented with a body temperature > 38 °C. Infants with an OAI of the hand due to *K. kingae* usually present with symptomatology suggestive of a musculoskeletal disease with either limpness or local swelling, with or without redness. The signs discernable upon physical examination will thus revolve around localized pain and swelling and a limited articular range of motion. The absence of a higher-than-normal temperature is common, and, unfortunately, this parameter has no predictive value when trying to exclude an infection.

The present study shows that OAIs of the hand and wrist caused by *K. kingae* are characterized by mild inflammatory responses. Thus, many treating physicians may often consider this non-suggestive of an OAI. Indeed, in the present cohort, the mean CRP value reached 28.3 mg/L, but was below 20 mg/L in 53.3% of children with a hand infection caused by *K. kingae*. The noted increase in CRP value was considered modest and relatively unspecific. Indeed, it is recognized that viral infections may give rise to moderate increases in CRP values, usually of up to 25 mg/L, whereas CRP levels can even exceed the threshold of 50 mg/L during rheumatoid arthritis. Even if this series of hand and wrist infections due to *K. kingae* is relatively small, the values revealed matched those published in rare previous studies [22]. Indeed, CRP levels reached 32 mg/L in a small series of patients with septic tenosynovitis caused by *K. kingae,* published by El Houmami et al. [22]. Similarly, Lironi et al. studied a small cohort of infants with tenosynovitis due to *K. kingae* and demonstrated that CRP levels averaged 39.9 mg/L at admission [19]. Thus, less than half of the infant patients with hand or wrist infections caused by *K. kingae* had a positive value for CRP, a predictor added by Caird et al. for improving the power of the Kocher algorithm for recognizing OAIs [33]. 

Our study also highlighted the caution required when interpreting ESRs during hand or wrist infections caused by *K. kingae*. Indeed, interpreting the ESR is questionable and should be done with circumspection. A total of 79.2% of the cases in our series had an ESR above 20 mm/h. However, when ESRs were interpreted with reference to the cut-off considered a predictor by Kocher et al. (ESR ≥ 40 mm/h), we noted that only 41.7% of patients with a hand or wrist OAI caused by *K. kingae* were above this limit. The ESR is a commonly used hematology test that can reveal an increase in inflammatory activity within the body caused by one or more conditions, such as autoimmune disease, infections, or tumors. The ESR is not specific for any one disease but is used in combination with other tests to determine the presence of increased inflammatory activity. Usually, bacterial infections are associated with an extremely high ESR (>100 mm/h). The ESRs recorded in this study, therefore, were only slightly or very rarely suggestive of a bacterial infection. Furthermore, it was impossible for us to compare these ESRs with rates appearing in the literature, since this infectious problem, for this germ and in this age group, has rarely been reported.

As is normally the case in children under 4 years old with an OAI due to *K. kingae*, WBC counts were only slightly elevated, and the mean WBC count reached 12,600/mm^3^: only 45.2% of children with a hand or wrist infection had a WBC count above 12,000/mm^3^. This is all the more significant as WBC counts < 17,500/mm^3^ in children 6–12 months old, <17,000/mm^3^ in children 13–24 months old, and <14,500/mm^3^ in children 25–48 months old are considered normal. Thus, it is clear that WBC count is a nonspecific index for inflammation, that it can be normal in as many as 80% of cases, and that its low sensitivity makes it an unreliable indicator of OAIs [41]. Thus, it appears that WBC count is a very poor marker of a *K. kingae* infection, and all the more so when these values are indexed to the normative values established for young children.

The present study’s findings showed that the platelet count was an interesting biological parameter for detecting and recognizing a hand or wrist OAI due to *K. kingae*. Indeed, our results showed that 45.5% of OAIs due to *K. kingae* were characterized by an increase in the platelet count, with a mean value reaching 401,000/mm^3^. Furthermore, 57.6% of the platelet counts measured were above the previously mentioned threshold of 361,500/mm^3^ described to differentiate between a pyogenic OAI and an OAI due to *K. kingae* [1]. In addition to the fact that definitions of thrombocytosis vary between >400,000/mm^3^ and >500,000/mm^3^, it is currently recognized that those anucleate cells also contribute to immune and systematic inflammatory processes. It has been shown that platelets can secrete pleiotropic immune and inflammatory mediators, which play crucial roles in the interactions with monocytes and neutrophils [42]. Thus, thrombocytosis may be reactive and secondary to an underlying inflammatory condition, such as tissue damage, malignancy, or infectious diseases [42]. Thrombopoiesis is usually inhibited after an acute bacterial infection, whereas, in contrast, chronic inflammation is often associated with reactive thrombocytosis [42,43,44]. Surprisingly, reactive thrombocytosis was more often associated with *K.* kingae OAIs than with methicillin-resistant *Staphylococcus aureus* infections [45], and this phenomenon is probably explained by the fact that OAIs caused by *K.* kingae generally follow a mild clinical course, usually behaving like a long-standing infection, which can lead to a more sustained reactive thrombocytosis.

The present study revealed that most OAIs of the hand or wrist caused by *K. kingae* are susceptible to medical treatment and that surgical treatment is more the exception than the rule. Even in cases involving highly significant infections of the tendon sheaths, *K. kingae* responds well to antibiotic treatment, and the tissue destruction seen with other, more aggressive pathogens, such as *Staphylococcus aureus* or beta-hemolytic *Streptococcus*, is rare. In the absence of severe clinical symptoms, antibiotic treatment can usually be initiated without surgical irrigation, even in cases of tenosynovitis due to *K. kingae*. However, because increased pressure in the tendon sheath can compromise vascular supply to the tendons, we believe that in cases of significant swelling, tendon sheath irrigation may constitute a necessary therapeutic alternative, especially if nucleic acid amplification assays are unavailable. Similarly, patients with septic arthritis of the hand or wrist caused by *K. kingae* could also be treated medically. It is clear, however, that in all cases of suspected *K. kingae* OAIs, it is essential to evaluate the evolution of this treatment modality closely, and in the event of a negative evolution, to suspect the existence of a pyogenic pathogen.

## 5. Conclusions

The present study’s findings lead us to believe firmly that most osteoarticular infections of the hand or wrist among children aged between 6 months and 4 years old are due to *K. kingae*. Hand and wrist infections due to *K. kingae* seem to be able to diffuse, insofar as they can affect several anatomical compartments, and arthritis and tenosynovitis are the most frequently diagnosed lesions. Like most *K. kingae*-associated infections, the hand or wrist involvement is often characterized by mild clinical picture and contained blood inflammatory markers, with the consequence that young patients present few, if any, of the typical criteria evocative of an OAI. Finally, most OAIs of the hand or wrist caused by *K. kingae* are susceptible to medical treatment, and surgical treatment is the exception rather than the rule.

## Figures and Tables

**Figure 1 microorganisms-11-02123-f001:**
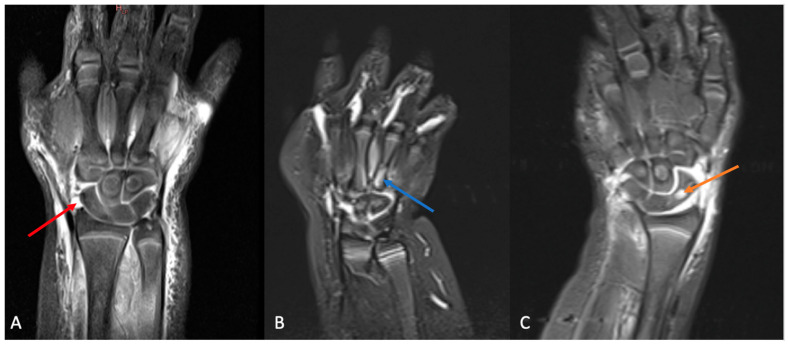
(**A**): T1 sequence MRI of a right wrist showing septic arthritis of the radiocarpal joint (red arrow) associated with cellulitis; (**B**): T1 sequence MRI of a right wrist showing septic osteomyelitis of the fourth metacarpal bone (blue arrow); (**C**): T1 sequence MRI of a left wrist showing subacute osteomyelitis of the scaphoid (orange arrow) associated with septic arthritis.

**Figure 2 microorganisms-11-02123-f002:**
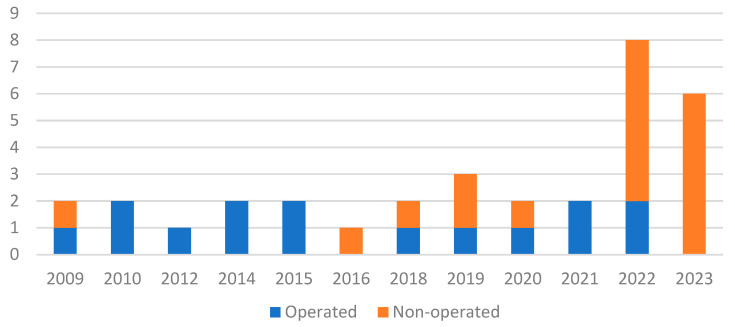
Distribution of therapeutic strategies by year.

**Table 1 microorganisms-11-02123-t001:** Demographic data.

Sex Ratio (F/M)	19/14
Age (months) Mean Standard deviation Minimum Maximum	16.9 9.4 7 48

**Table 2 microorganisms-11-02123-t002:** Distribution of diagnoses.

Diagnostic	Counts	% of Total
Septic arthritis	23	69.7%
Tenosynovitis	17	51.2%
Osteomyelitis	10	30.3%
Cellulitis	10	30.3%
Myositis	4	12.1%

**Table 3 microorganisms-11-02123-t003:** Distribution of compartment involvement.

		Number	% of Total
Joint involvement	Yes No	23 10	69.7% 30.3%
Bone involvement	Yes No	10 23	30.3% 69.7%
Soft tissue involvement	Yes No	23 10	69.7% 30.3%

**Table 4 microorganisms-11-02123-t004:** Distributions of the clinical and laboratory parameters.

	Temperature (°C)	WBC (G/L)	Left Shift (%)	PLT (G/L)	CRP (mg/L)	ESR (mm/h)
N	33	33	30	33	32	26
Missing	0	0	3	0	1	7
Mean	37.4	12.5	1.83	401	31.0	40.9
Median	37.1	11.9	0.00	380	20.0	38.0
Standard deviation	0.960	4.37	4.54	131	29.9	19.2
Interquantile range	1.2	6.4	2.0	158	32.3	26.3
Minimum	35.9	4.40	0.00	123	2	2
Maximum	40.2	20.6	24.5	704	115	74

WBC = white blood cell; PLT = platelet count; CRP = C-reactive protein; ESR = erythrocyte sedimentation rate.

## Data Availability

Data are available upon reasonable request.

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
