# Peer review of "Pediatric Osteoarticular *Kingella kingae* Infections of the Hand and Wrist: A Retrospective Study"

_microorganisms, 2023, doi:10.3390/microorganisms11082123_

Round 1

Reviewer 1 Report

This study aimed to examine the pediatric osteoarticular infection in hand and wrist by kingella kingae through a large patient cohort. From included 33 patients, the authors found that the infections could even diffuse into several anatomical compartments causing arthritis and tenosynovitis. In general, this study is of interest and matching the scope of the journal. However, there are a few comments need to be addressed. 

1. The authors have included 33 patients from a large cohort and extracted information on age, sex, temperature at admission, the bone, joint, or tendon and so on. However, it is not presented very well so that the readers could not get an overview of these patients. I would suggest the authors to add a new table to present the baseline of all the included patients. 

2. In Fig.1, please mark the infected compartment to make it more visible since the authors have mentioned several compartments could be infected.

3. The authors discussed different parameter for the diagnosis for OAI. Please give more details about the MRI parameters for the diagnosis of OAI. Furthermore, discuss more about the MRI chracterastics of the kingella kingae infected OAI and distinguish it with other OAIs, which would be helpful for the future diagnosis. 

4. The authors observed that only 15.2% had a temperature above 38.5°C during their clinical examination at admission. Do the authors get some clue how to diagnose this type of patients at admission especially without a clear fever from this study? Please describe it in details. 

5. The authors described that 54.5% underwent medical 178 management with an appropriate antibiotic therapy, whereas 45.5% underwent surgical 179 treatment. I would is there any difference between these two treatments? Furthermore, do the authors have the recommendation for how to choose the treatments for these patients from this study? Please clarify it. 

Reviewer 2 Report

In this paper authors described K. kingae infections of the hand and wrist, that are less common presentation than other bone and joint location. They identified 33 cases (14 proven and 19 highly probable) from their large cohort.

I have a few comments:

- it is a bit confusing that percentages in Table 1 were calculated from the total number of diagnoses (n=64) instead of the total number of patients (n=33). Indeed, if 23 arthritis were diagnosed, it seems more interesting to show that artritis was diagnosed in 23/33=69.7% patients that in 23/64=35.9% of all kind of diagnosis

- Figure 1: it would be good to add some arrows on the figures to show the lesions

- Table 3: please add the interquartile ranges

- p5 lines178-181, please add how many patients with septic arthritis and tenosynovitis were medically managed with no surgery

- lines 298-308 in the discussion section, authors discussed about medical and surgical treatment in case of tendons involvement. It would be of interest to discuss also the strategy in case of joint involvement 

- p6 line 209 about the prevalence of oropharyngeal carriage, authors should cite the papers from Israel and France, and not only from Switzerland

-lines 239 and 314: please write K. kingae in italics, as well as in the references

Reviewer 3 Report

Dear Authors,

In this review, ,, Pediatric osteoarticular Kingella kingae infections of the hand and wrist: a retrospective study,, by Blaise Cochard et al. , aimed to examine osteoarticular infections of the hands and wrists caused by Kingella kingae in a large cohort of patients, to explore the implications of K. kingae infections of the hands or wrists based on MRI scans, to summarize and critically review the different types of lesions encountered in osteoarticular infections of the hand and wrist due to K. kingae.

Comments and Suggestions for Authors

The introduction gives us enough background, it includes relevant references.

The methods are adequately described.

The results are presented and discussed very well.

The figure and tables are explained in detail.

The final conclusions are pertinent.

The manuscript is not overloaded with unnecessary information.

Most of the bibliographic references are recent, and adequate to related and previous work.

The article has some shortcomings:

-      Authors should double-check abbreviations and make the necessary corrections so that abbreviations are explained when they first appear, both in the abstract and in the manuscript text and figure legends;

-      Double-check references and make necessary corrections; References must be numbered in order of appearance in the text (including citations in tables and legends)

-      in line 38 you specified the references [16,17], then in line 40, you jump to reference 23 [12,23,24].

-      The authors must specify which of the parameters in Table 3 had a normal distribution and are presented as the mean accompanied by the SD and which did not have a normal distribution and are presented as the median.

Moreover, the authors must specify in 2.2. Statistical Analysis, if they tested normality and with what test, tests.

-      References 18-22 are not found in the manuscript,

-      Authors must insert the reference, Caird et al., presented in line 253

Overall Recommendation: Accept after minor revision.

Minor editing of the English language required
